# Treatment of a Recurrent Pyometra by Surgical Uterine Drainage in a Main Coon Cat

**DOI:** 10.3390/vetsci10010060

**Published:** 2023-01-15

**Authors:** Gianluca Martini, Roberta Bucci, Salvatore Parrillo, Augusto Carluccio, Maria Carmela Pisu

**Affiliations:** 1Clinica Veterinaria Borghesiana, 00133 Roma, Italy; 2Department of Veterinary Medicine, University of Teramo, Piano d’Accio, 64100 Teramo, Italy; 3VRC—Centro di Referenza Veterinario, 10138 Torino, Italy

**Keywords:** feline reproduction, fertility, pyometra, conservative treatment, uterine drainage

## Abstract

**Simple Summary:**

A 3-year-old Main Coon female cat was referred for recurrent pyometra, previously treated with aglepristone and cloprostenol. Due to its high reproductive value, ovariohysterectomy was not an option so, being the medical treatment ineffective, a uterine drainage and flushing were performed. Under general anesthesia, the patient underwent a laparotomy; a sterile urinary catheter was inserted in the uterine horns and 150 mL of milky and thick liquid was collected. A lavage with lukewarm sterile saline was also associated. After surgery, aglepristone and antibiotics were administered. The cat showed no recurrence six months after surgery and had an uneventful pregnancy.

**Abstract:**

Pyometra is a uterine disease typical of the luteal phase of the estrus cycle. For selected patients, such as breeding subjects, ovariohysterectomy is not a valid resolutive option. Medical treatments involving cloprostenol and aglepristone have been developed for the cats, but they can be ineffective in rare cases. Transcervical drainage and flushing have been described for the dogs, as well as for large wild cats. However, to the author’s knowledge, there are no report of uterine drainage in cats. The present case describes an alternative treatment of pyometra in a 3-year-old Main Coon previously treated with aglepristone. The patient underwent a laparotomy: the uterus was exposed, and a sterile urinary catheter was inserted into each horn, through the wall of the uterus, to allow the drainage of pathological collection and a subsequent lavage with lukewarm sterile saline. Medical treatment with aglepristone and marbofloxacin was associated. After treatment, no recurrence was reported, and the cat had an uneventful pregnancy. Although it is a unique case report, the results presented are promising, as the technique appears to have provided healing and preserved fertility. Further studies are needed to confirm its efficacy in the long-term prevention of recurrence.

## 1. Introduction

Pyometra is a potentially life-threatening uterine disease, affecting intact female mature cats [1,2], caused by an opportunistic bacterial infection (mainly *E. coli*) [3].

In cats, pyometra often progresses in a mild form with few symptoms [2], especially in young animals [4]; a thick or sero-hemorrhagic vaginal discharge can be present, usually hidden by the cats’ cleaning habits [5,6]. Vomiting, lethargy, and anorexia are other reported, but non-specific, symptoms [7]. Unlike dogs, polyuria and polydipsia are not commonly seen in cats [2]. 

Traditionally, the ovariohysterectomy was considered elective for the treatment of pyometra [2,5,6,8,9], but it is not a valid option for patients intended for reproduction [10]. Consequently, various medical treatments have been developed [2,6,11]. The most used drugs are synthetic analogs of prostaglandins, such as cloprostenol, [4,11] and aglepristone, an antiprogestin [7,10]. Cloprostenol (5 µg/kg) can be administered for three consecutive days with mild and transient side effects, such as nausea, vomiting, and abdominal pain [11]. The use of aglepristone, on the other hand, does not show side effects; the classic administration protocol provides for its use on days 1, 2, and 7 [10], in association [5] or not [7] with cloprostenol. A modified protocol has also recently been proposed, involving the use of aglepristone on days 1, 3, 6 and 9 without associating prostaglandins [12]. In cats, unlike dogs, aglepristone should be administered at a higher dose (15 mg/kg) due to its different pharmacokinetics in the feline species [13], as it binds progesterone receptors with an affinity nine times higher than the endogenous hormone, decreasing intrauterine progesterone concentration, but its bioavailability is reduced [12,13].

In all cases, using broad-spectrum antibiotics, such as amoxicillin-clavulanic acid and trimethoprim-sulfadoxine [14] is associated with both medical and surgical treatment [6]. According to more recent reports, fluoroquinolones have also been used, following the emergence of resistance to broad-spectrum antibiotics [12,15,16].

In some cases, medical treatment with aglepristone may not be effective. A case series by Nak et al. [7] reports a success rate of 90% in cats treated with this progesterone-receptor blocker administered in days 1, 2, 7 and 14 [7]. For dogs, recovery rates after medical treatment also ranges from 46 to 100% [5]. 

Another therapeutic option reported for treating pyometra and preserving fertility is a local drug administration or uterine drainage. Even if these options are not commonly used, there are some reports of usage in dogs. Gabor et al. infused prostaglandin F2α into the vaginal lumen [17], while Lagerstedt et al. developed a device for transcervical draining and flushing [18,19,20]. More recently, a similar device has been described by Gurbulak et al. for the transcervical administration of antibiotics in bitches also treated with aglepristone [21]. Additionally, in 2010, De Cramer described a laparotomy-assisted transcervical drainage and lavage in dogs with pyometra [22].

To the author’s knowledge, there are no report of similar treatments for domestic cats, but Hildebrandt et al. reported the use of a nonsurgical uterine lavage technique for the treatment of uterine infection-induced infertility in large felids [23].

The present report describes the association of uterine drainage and lavage and medical treatment, to improve its efficacy and shorten duration, in a cat affected by relapsing pyometra.

## 2. History and Clinical Findings

A 3-year-old intact Main Coon female cat was referred for a gynecological examination following the occasional finding, during an ultrasound examination, of an increase in the size of the uterus.

Reproductive history reported no previous pregnancy, but two episodes of pyometra, treated with aglepristone, administered on day 1 and day 2, cloprostenol, administered in days 3 to 5, and amoxicillin-clavulanic acid for 7 days. No vomiting or lethargy was reported, and the owners only reported a mild inappetence.

During a thorough clinical examination, the patient was in good general condition and the evaluation only detected a mild vulvar discharge. Hematological and serum biochemical tests showed no alteration. 

The stage of the reproductive cycle was also investigated. The vaginal cytological smear showed a percentage of keratinized cells higher than 80%, considered indicative of estrus [24]; the serum progesterone dosage was less than 1 ng/mL (in cats, values higher than 1 ng/mL are indicative of ovulation [25]). Moreover, the cat showed estrous behavior (lordosis posturing and tail deviation [25]) in response to the stimulation of the back and the perineal area.

An ultrasonographic (US) examination of the genital tract, performed at the time of admission with a 10.0 MHz linear probe (MyLab Alpha, Esaote S.p.A. Italia), revealed a significant uterine enlargement: the maximum uterine diameter recorded was 26.2 mm and the lumen was filled with non–homogeneous, corpuscular material. In addition, a slight periuterine effusion was found (Figure 1).

The US appearance was evocative of a uterine infection, consistent with the previous history of recurrent pyometra.

## 3. Treatment and Follow up

In consultation with the owners, who strongly wished to preserve the animal’s fertility, the patient underwent medical treatment again for the resolution of the uterine infection.

Ovulation was induced with 250 UI of Human Chorionic Gonadotropin (HCG) (Corulon^®^, MSD Animal Health S.r.l., Rahway, NJ, USA) and the cat was hospitalized. After 48 h, progesteronemia was 3.45 ng/mL, confirming the luteal stage [25]. Medical treatment for pyometra was then started. A subcutaneous injection of 15 mg/kg of Aglepristone (Alizin^®^; Virbac, Carros, France) was administered on Day 1 (D1), D2, and D8, followed by subcutaneous administration of 1.5 mcg/kg of cloprostenol (Dalmazin^®^, Fatro, Bologna, Italy) in days 3 to 6 (D3–D6) [2]. An antibiotic therapy with 3 mg/kg of marbofloxacin *per os* (Marbocyl^®^; Vetoquinol S.r.l., Bertinoro, Italy) for 7 days [12] was associated.

Subsequent US evaluations performed at D5 and D8 showed a normal uterus with no intraluminal collection.

Three days after the end of the treatment, the cat showed a new heat behavior and mated spontaneously with a cohabiting cat. An ultrasound examination was performed on day 21 after mating; it did not confirm pregnancy but revealed recurrence of fluid accumulation in the uterus (Figure 2).

Since previous medical therapies were unsuccessful, surgical drainage of the uterus was performed, along with re-administration of aglepristone, to preserve fertility.

The patient was prepared for a laparotomy and the uterus was externalized (Figure 3); a sterile urinary catheter (urinary catheter Arnolds 1 mm × 30.5 cm, Portex^®^, Smiths Medical, Weston, MA, USA) (Figure 4) was inserted into the uterus through a 2 mm incision in the apex of each horn (Figure 4 and Figure 5). The pathological accumulation was collected using first a syringe, to take samples for subsequent cytological and bacteriological examination, and then a surgical aspirator.

A total of 150 mL of milky and thick liquid (Figure 6) was collected, then a sterile swab was carried out on the endometrium in search of pathogens. Finally, uterine lavage was performed with 15 mL of lukewarm sterile saline solution, and both the uterine and the abdominal surgical wounds were closed. In addition, 15 mg/kg of aglepristone was administered subcutaneously for two consecutive days after the surgery, and 3 mg/kg of marbofloxacin for 7 days was administered *per os* [12].

The post-operative period was uneventful. US evaluation performed three days after treatment did not reveal pathological fluid accumulation (Figure 7).

Cytology on the collected fluid revealed numerous segmented neutrophils, thus confirming the diagnosis of pyometra. Bacterial cultures on both samples were instead negative. 

Seven days after surgery, the patient was finally discharged without further treatment.

Regular US checks for six months ruled out further recurrence of pyometra, so the cat was naturally mated with a healthy male and had an uneventful pregnancy.

## 4. Discussion

Uterine lavage and intrauterine drugs administration are commonly used in equine medicine, to treat endometritis [26]. In the mare, transcervical access is easy to perform due to the size of the animal. In small animals practice, this approach is more difficult due to the smaller size of the uterus, but it has been reported in bitches. Some authors have developed a modified transcervical catheter that allows grasping of the cervix and subsequent uterine drainage or drugs infusion [18,19,20]. Similar techniques have also been used in large wild felids to treat uterine infection-induced infertility [23].

More recently these procedures are uncommon; in small animals, pyometra is mainly treated with medical treatments [2,5]. In rare cases, however, these procedures are ineffective [5,7] and there is a need for alternative solutions for patients who cannot be spayed. If a transcervical approach can be attempted in large-sized dogs, as described in various articles [18,19,20,21,22], for cats, the relatively small size (even in large breeds) makes this technique difficult to perform.

The transcervical approach has been described, in cats, to perform artificial insemination. Zambelli et al. have, in fact, developed a modified urinary catheter to perform transcervical inseminations, with laparoscopic assistance [27,28]. A similar technique could not be used in the present case, since at the time of surgery the cat was in diestrus, and the cervix was closed [29]. A pharmacological approach, allowing the opening of the cervix, would have been possible but would have required more time. Instead, the chosen surgical approach was immediate.

The authors decided to perform uterine drainage modifying the uterine horn insemination technique described for cats by Tsutsui et al. [30]. In the aforementioned article, an 18 G needle was used to inseminate the animals. In the present case, a sterile urinary catheter was preferred due to a minimal uterine injury, as well as to its flexibility and length, that allowed easy drainage of each horn.

The technique presented was also less invasive than the one described for dogs, in which the surgeon manipulated the uterus to allow the transcervical catheter to pass through the cervix [22].

Furthermore, the laparotomic approach allowed operators to accurately verify that pus or lavage fluids were not being forced into the abdomen through the fallopian tubes, as also described by De Cramer [22].

As for the patient, in the authors’ opinion, the first recrudescence of pyometra, reported in medical history, was related to the incorrect administration of aglepristone. In fact, even when administered together with prostaglandins, another administration on day 8 should be performed if the uterus is not completely empty [2,7]. Unfortunately, in the present case, even the administration of aglepristone with the correct and complete protocol did not prevent a further recurrence. This occurrence determined the authors’ choice to treat the new recurrence by associating drainage and surgical lavage with medical therapy.

In cases of feline pyometra there is an increasing incidence of broad-spectrum antibiotic resistance [12], that could support the unsatisfactory outcome of the previous medical treatment attempted, involving the use of amoxicillin and clavulanic acid association. This is also the reason why, after surgery, a third-generation fluoroquinolone was preferred, as also reported by some authors in cases of pyometra [12,15,16]. In the authors’ experience, unfortunately, this antibiotic class is often the only effective in the treatment of pyometra. Moreover, using broad-spectrum antibiotics while awaiting the susceptibility test and then moving on to more effective molecules resulted in prolonged therapeutic times, to the detriment of the patients. 

Regarding bacteriological tests performed on samples collected during surgery, the absence of bacterial growth could be related to the inhibiting effect of activated neutrophils on the bacteria. Even in equine practice, there is sometimes a mismatch between bacterial cultures and evidence of inflammatory cells. In these cases, cytological evaluation is considered the gold standard [31]. Furthermore, in mares with endometritis, several uterine lavages are required to restore sterility [32]. In the present case only one lavage was performed. The risk of new bacterial growth was reduced by combining surgical treatment with medical treatment for pyometra using aglepristone and antibiotics.

After surgery, a short protocol with aglepristone administrated in days 1 and 2, and marbofloxacin was associated. Subsequent ultrasound examinations did not reveal any recrudescence and no further treatment was needed. This finding confirmed, also for cats, De Cramer’s hypothesis that the association of uterine drainage was effective in shortening the medical treatment of pyometra [22]. Finally, no short-term recurrences were detected, and the cat had an uneventful pregnancy, confirming the effectiveness of the proposed treatment in restoring a healthy uterine environment.

## 5. Conclusions

To the authors’ knowledge, this is the first time that uterine drainage and lavage have been described for cats. Even knowing that it is a unique case report, the results are promising. In fact, the association of medical treatment and surgical drainage of the uterus has proved to be effective in prevention of short-term recurrence of pyometra and in preserving fertility. Further studies are needed to verify the long-term efficacy of the proposed technique in recurrence prevention. If confirmed, it could provide another useful protocol for the treatment of pyometra in breeding cats.

## Figures and Tables

**Figure 1 vetsci-10-00060-f001:**
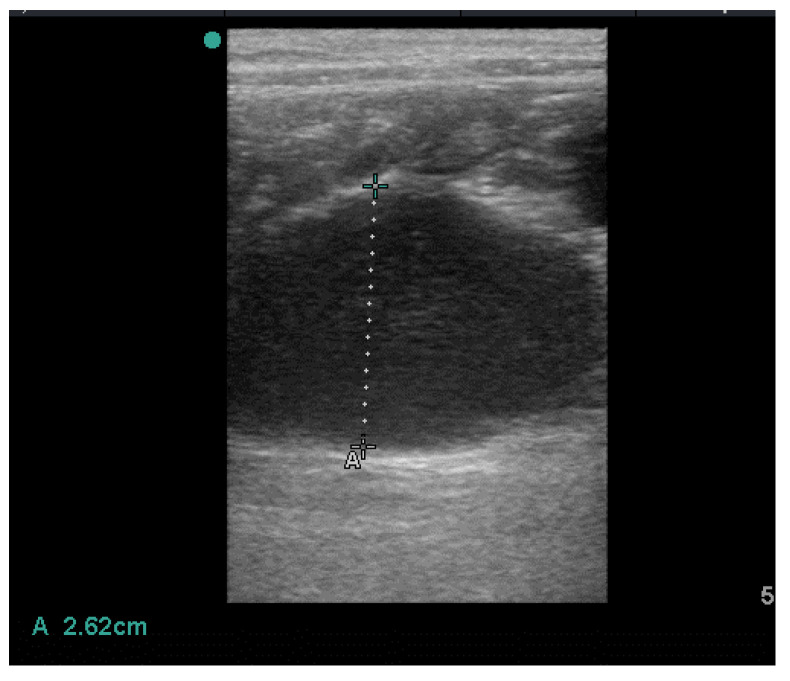
Uterine collection.

**Figure 2 vetsci-10-00060-f002:**
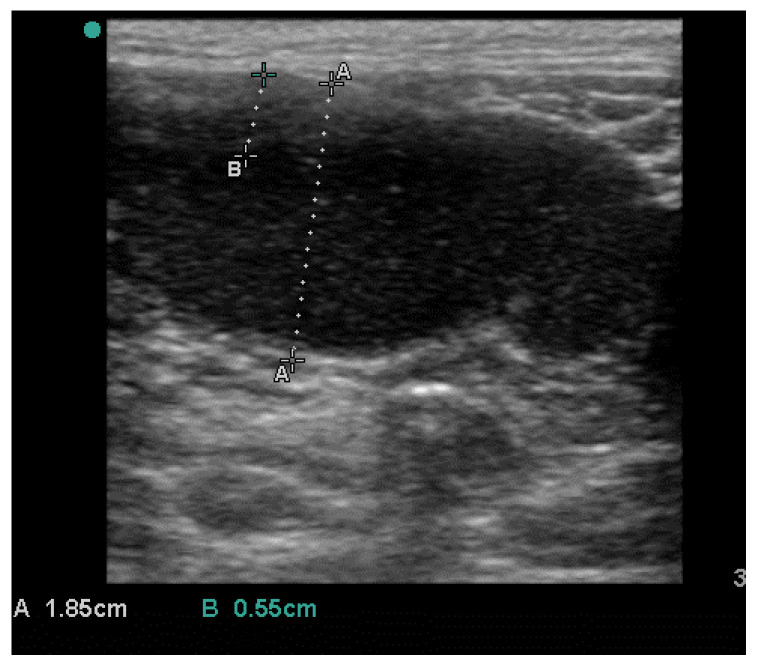
Recurrence of pyometra after medical treatment.

**Figure 3 vetsci-10-00060-f003:**
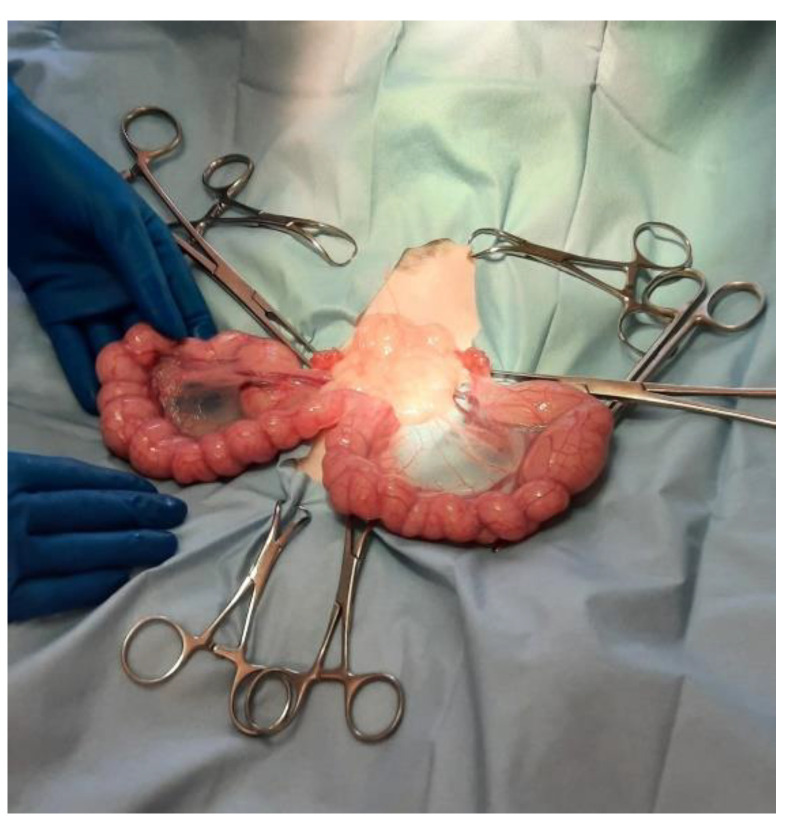
The uterus is externalized.

**Figure 4 vetsci-10-00060-f004:**
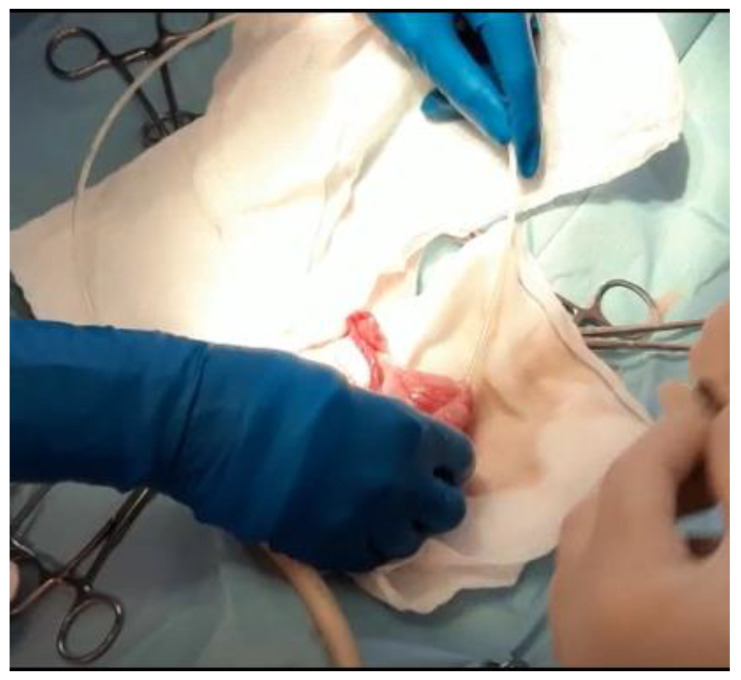
A sterile catheter is inserted into the uterus through a 2 mm incision in the apex of each horn.

**Figure 5 vetsci-10-00060-f005:**
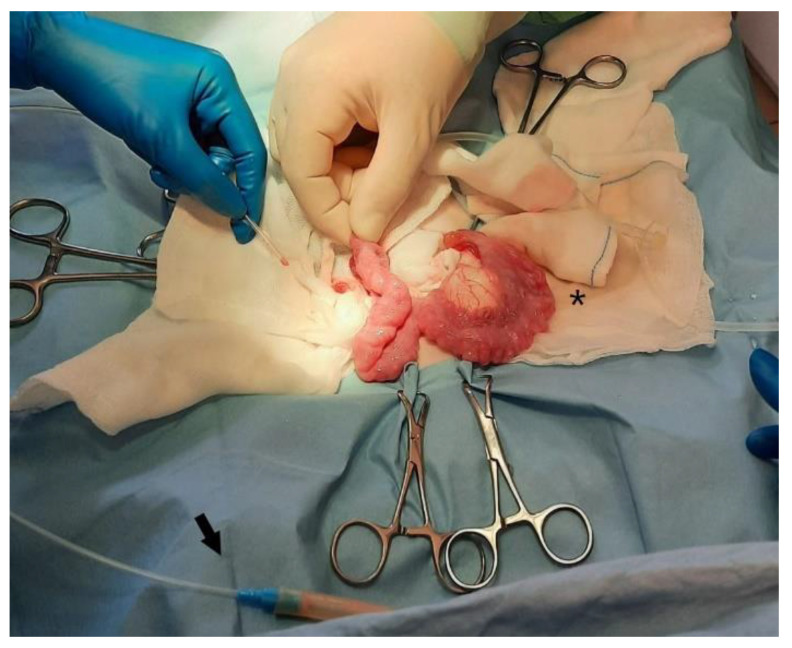
Detail of the size difference between left drained horn (*), and right, still to be drained. The arrow indicates the connection between the catheter and the surgical aspirator.

**Figure 6 vetsci-10-00060-f006:**
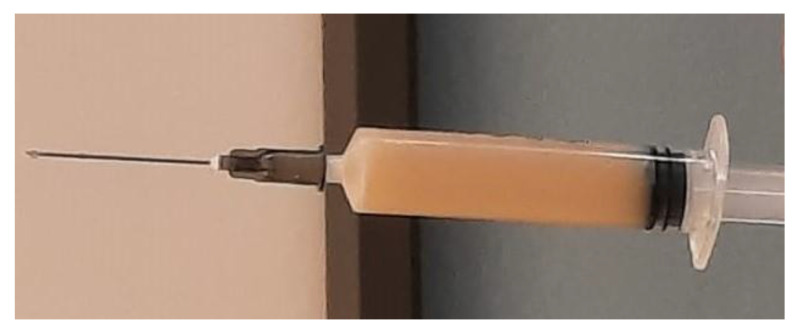
Sample of the fluid drained from the uterus.

**Figure 7 vetsci-10-00060-f007:**
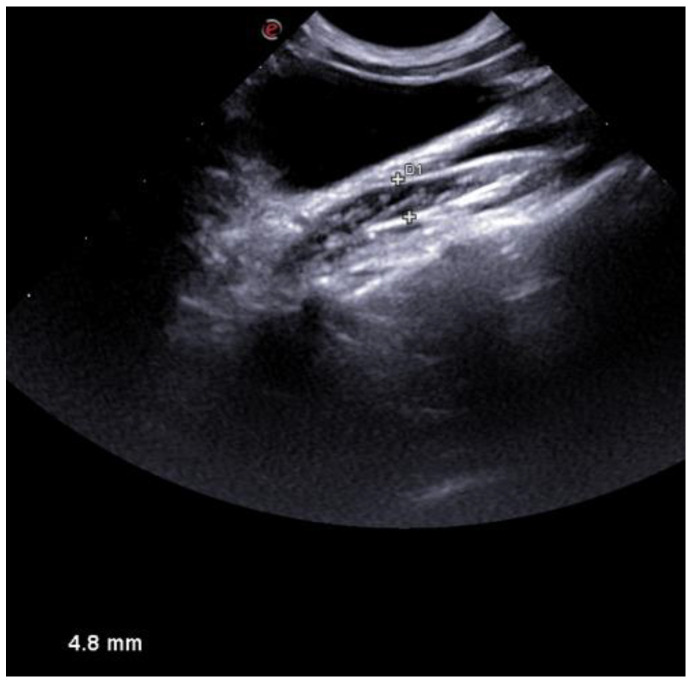
US appearance of the uterus three days after treatment. No collection was detected.

## Data Availability

The data presented in this study are available on request from the corresponding author.

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
