# Peer review of "Treatment of a Recurrent Pyometra by Surgical Uterine Drainage in a Main Coon Cat"

_vetsci, 2023, doi:10.3390/vetsci10010060_

Round 1
Reviewer 1 Report
The manuscript entitled “Treatment of a recurrent pyometra in a 3-year-old Main Coon female cat by surgical uterine drainage” describes a case report of a cat treated successfully for pyometra. However, the manuscript provide a poor description and details regarding the procedures. Unfortunately, the introduction section lacks a strong background to support pyometra treatment. Authors should provide a stronger background regarding importance as aspects of feline pyometra and support the ideal of a clinical treatment. In the case description (divided into two topics) the details informed are very short not allowing replication. Regarding treatment, no other researcher will be able to reproduce the treatment based on the author’s description. In conclusion, the two first phrases are not conclusions.
Author Response
Manuscript ID: Vetsci-2117884
Response to Review (Round 1)
To Editor and Reviewers,
We would like to thank you for the opportunity to revise our paper “Treatment of a recurrent pyometra in a 3-year-old Main Coon female cat by surgical uterine drainage”. We corrected the manuscript based on the comments received.
All revisions have been added to the formatted manuscript provided by the Editors and highlighted to be easily viewed by the Editors and Reviewers. Moreover, the manuscript has been carefully checked by a native English colleague (English revisions are highlighted, as well). All Authors approved the content of the submitted manuscript.
Please see below, in italics, for a point-by-point response to the Editor’s and Reviewers’ comments and concerns. All line numbers refer to the revised manuscript file.
Reviewer 1
The manuscript entitled “Treatment of a recurrent pyometra in a 3-year-old Main Coon female cat by surgical uterine drainage” describes a case report of a cat treated successfully for pyometra. However, the manuscript provides a poor description and details regarding the procedures. Unfortunately, the introduction section lacks a strong background to support pyometra treatment. Authors should provide a stronger background regarding importance as aspects of feline pyometra and support the ideal of a clinical treatment. In the case description (divided into two topics) the details informed are very short not allowing replication. Regarding treatment, no other researcher will be able to reproduce the treatment based on the author’s description. In conclusion, the two first phrases are not conclusions.
Thank you for your suggestions that help us improve our manuscript.
The introduction has been revised and information have been added regarding the clinical aspects and medical treatment of feline pyometra, as suggested. Moreover, in the authors’ opinion, for patient intended for reproduction, medical treatment is preferable. When it is not resolutive, an association of medical systemic treatment and local treatment (draining or pharmacological) can be attempted to improve effectiveness of treatment and shortener the duration of the therapies. Please, see lines 40-42, 47-57 of the corrected ms.
In addition, the case description has been improved by adding missing information about the procedure and images to explain better all the procedures performed (Lines 137-143).
Reviewer 2 Report
The manuscript presents the case report about the use surgical uterine drainage for the supportive treatment of recurrent pyometra in a cat. As uterine lavage is barely used in feline medicine, this case report shows an interesting option which can be added to conventional treatment.
The manuscript is properly structured and generally easy to follow. The introduction part provides sufficient background. The description of the methodology of uterine drainage is appropriate and enable the readers to repeat the procedure, however there are some information missing regarding other assessments (see below). Several pictures enrich the manuscript.
I have one major concern about the interpretation of this case: the authors have written that a bacteriological swab was taken after drainage of the content, but before uterine lavage (Line 123) and the result of bacteriological culture was negative. In the discussion it was written (Line 188) that ‘Additionally, uterine swabs after lavage confirmed that the procedure was effective in restoring uterine sterility’. Besides this discrepancy (before/after lavage), I do not believe that single flushing with physiological saline would make the uterus completely sterile. In horses it takes several lavages to eliminate the bacteria (e.g. DOI: 10.21836/PEM19970516). The negative result may be due to the technique – again, in mares it is well known that samples obtained by swabs can be false negative (comparing to samples obtained by uterine flushing).
Additionally, authors wrote that (Line 115) ‘The uterine fluid was collected using first a syringe, to take samples for subsequent bacteriological examination, and then a surgical aspirator’, but the results of this bacteriological culture are not provided. Was it negative similarly to the swab? If yes, what about the idea that it was not pyometra, but mucometra (not so uncommon in cats?). I think it should be considered and at least discussed in the manuscript.
Also, it is not stated (Line 101) which particular antibiotic was used during first treatment attempt. If it was not a marbofloxacin, therefore it is not sure if success of the treatment was due to the uterine drainage or because of changing the antibiotic for more effective one.
To sum up, in my opinion, stating that ‘the technique has proved to be effective in restoring uterine sterility and in preserving fertility’ is an exaggeration. However, I do agree that uterine drainage can be a useful supportive treatment in complicated cases and can (should?) be described as such.
I have some minor notes listed below.
I believe that after a addressing major concerns and after minor revision this manuscript will be suitable for publication.
Minor remarks:
- Line 60 and further: using the phrase ‘large cats’ without contexts may mislead the reader, suggesting large domestic cat breed e.g. Main Coon. I would write ‘large wild cats’ or more specifically: ‘tigers and leopards’.
- Line 61: ‘a trans uterine drainage’- better just ‘uterine drainage’ or ‘surgical uterine drainage’. Word ‘trans’ suggest going across sth, in this case catheter was inserted into the uterine lumen.
- Line 106: information could be added when the US for diagnosis confirmation was performed (which day after mating)
- Line 132: which days was the US performed?
- Line 133 and further: instead of ‘uterine collection’, better ‘uterine content’, fluid accumulation’ etc.
- Line 144: ‘due to the wide size ranges of the patients’, better ‘due to the small size of the patients’.
Author Response
Manuscript ID: Vetsci-2117884
Response to Review (Round 1)
To Editor and Reviewers,
We would like to thank you for the opportunity to revise our paper “Treatment of a recurrent pyometra in a 3-year-old Main Coon female cat by surgical uterine drainage”. We corrected the manuscript based on the comments received.
All revisions have been added to the formatted manuscript provided by the Editors and highlighted to be easily viewed by the Editors and Reviewers. Moreover, the manuscript has been carefully checked by a native English colleague (English revisions are highlighted, as well). All Authors approved the content of the submitted manuscript.
Please see below, in italics, for a point-by-point response to the Editor’s and Reviewers’ comments and concerns. All line numbers refer to the revised manuscript file.
Reviewer 2
The manuscript presents the case report about the use surgical uterine drainage for the supportive treatment of recurrent pyometra in a cat. As uterine lavage is barely used in feline medicine, this case report shows an interesting option which can be added to conventional treatment.
The manuscript is properly structured and generally easy to follow. The introduction part provides sufficient background. The description of the methodology of uterine drainage is appropriate and enable the readers to repeat the procedure, however there are some information missing regarding other assessments (see below). Several pictures enrich the manuscript.
Thank you for your review that help us improve our manuscript. During the revision of the manuscript, missing information have been added to the case description, along with other images (Lines 137-143). Please, see below for point-by-point response.
I have one major concern about the interpretation of this case: the authors have written that a bacteriological swab was taken after drainage of the content, but before uterine lavage (Line 123) and the result of bacteriological culture was negative. In the discussion it was written (Line 188) that ‘Additionally, uterine swabs after lavage confirmed that the procedure was effective in restoring uterine sterility’. Besides this discrepancy (before/after lavage), I do not believe that single flushing with physiological saline would make the uterus completely sterile. In horses it takes several lavages to eliminate the bacteria (e.g. DOI: 10.21836/PEM19970516). The negative result may be due to the technique – again, in mares it is well known that samples obtained by swabs can be false negative (comparing to samples obtained by uterine flushing). Additionally, authors wrote that (Line 115) ‘The uterine fluid was collected using first a syringe, to take samples for subsequent bacteriological examination, and then a surgical aspirator’, but the results of this bacteriological culture are not provided. Was it negative similarly to the swab? If yes, what about the idea that it was not pyometra, but mucometra (not so uncommon in cats?). I think it should be considered and at least discussed in the manuscript.
Thank you for your observation. As stated in “treatment and follow up” section, a first sample for cytology and bacteriology was obtained collecting the pus with a syringe. A second sample for further exam has been collected with a swab on the endometrium after having drained all pathological accumulation but, as you noted, before uterine lavage. No other swabs have been performed after lavage with sterile saline. The discrepancy you noted in the “discussion” section is due to an error during the translation process. I apologize for this.
Moreover, a cytological examination has been performed on drained fluids, evidencing the presence of segmented neutrophils, thus confirming the suspicion of pyometra (text has been integrated accordingly, please, see lines 157-159) and ruling out the idea of a recurrent mucometra.
Concerning the interpretation of bacteriology, I agree that the negative result could be related to a false negative, as for the horses (Nielsen, J. M. (2005). Endometritis in the mare: a diagnostic study comparing cultures from swab and biopsy. Theriogenology, 64(3), 510-518. https://doi.org/10.1016/j.theriogenology.2005.05.034 ) and a complete sterility needs several flushing to be achieved (Mattos, R., Castilho, L. F. F., Malschitzky, E., Naves, A. P., Keller, A., Gregory, R. M., & Mattos, R. C. (1997). Uterine lavage with saline in mares as treatment for endometritis. Pferdeheilkunde, 13, 521-524. doi: 10.21836/PEM19970516).
The negative results on culture could also be related to the known inhibiting activity of neutrophils on the bacterial growth; All considered, an antibiotic treatment was carried out in any case, even in the absence of evidence of bacterial growth.
Discussion section has been improved, accordingly, please see lines 207-218
Also, it is not stated (Line 101) which particular antibiotic was used during first treatment attempt. If it was not a marbofloxacin, therefore it is not sure if success of the treatment was due to the uterine drainage or because of changing the antibiotic for more effective one.
I apologize for the unclearness. Marbofloxacin was also used in the first therapeutic attempt by authors (correction has been made in lines 115-116). Based on recent literature and on our experience, and given the previous use of broad-spectrum antibiotics, fluoroquinolones appear to be more effective in treating uterine infections. In our opinion, the association of medical treatment (aglepristone and marbofloxacine) and uterine drainage has been the key for the resolution of this case. The ms has been corrected, please see lines 226-227
To sum up, in my opinion, stating that ‘the technique has proved to be effective in restoring uterine sterility and in preserving fertility’ is an exaggeration. However, I do agree that uterine drainage can be a useful supportive treatment in complicated cases and can (should?) be described as such.
Thank you for your suggestion. The statement has been modified as follows: “the association of medical treatment and surgical drainage of the uterus has proved to be effective in prevention of short-term recurrence of pyometra and in preserving fertility.” (Lines 226-227).
Abstract has been modified accordingly, as well. Please see lines 29-30
I have some minor notes listed below.
I believe that after addressing major concerns and after minor revision this manuscript will be suitable for publication.
Thank you for your consideration and suggestions.
Minor remarks:
- Line 60 and further: using the phrase ‘large cats’ without contexts may mislead the reader, suggesting large domestic cat breed e.g. Main Coon. I would write ‘large wild cats’ or more specifically: ‘tigers and leopards’.
Thank you, text has been corrected using “large wild cats” or “Large felids”
- Line 61: ‘a trans uterine drainage’- better just ‘uterine drainage’ or ‘surgical uterine drainage’. Word ‘trans’ suggest going across sth, in this case catheter was inserted into the uterine lumen.
Corrected, thank you
- Line 106: information could be added when the US for diagnosis confirmation was performed (which day after mating)
I apologize for the unclearness. US for pregnancy diagnosis was performed 21 days after mating. Ms has been corrected accordingly (please, see line 120).
- Line 132: which days was the US performed?
I apologize for the unclearness. US was performed 3 days after surgery. Ms has been corrected accordingly (Please, see line 153 and Figure 7).
- Line 133 and further: instead of ‘uterine collection’, better ‘uterine content’, fluid accumulation’ etc.
Corrected, thanks
- Line 144: ‘due to the wide size ranges of the patients’, better ‘due to the small size of the patients’
Corrected, thanks
Round 2
Reviewer 1 Report
The authors have improved the manuscript and this reviewer has no further comments.
Author Response
Authors are thankful to the Reviewer for their considerations that improved the overall quality of the manuscript.
Reviewer 2 Report
All the notes and issues has been addressed satisfactory, the manuscript has been improved and can pe published in present form.
Author Response

(The authors gave the same response as above.)
